

# Description of a new species of cryptic snubnose darter (Percidae: Etheostomatinae) endemic to north-central Mississippi

Ken A. Sterling and  Melvin L. Warren, Jr.

USDA Forest Service, Southern Research Station, Stream Ecology Laboratory, Oxford, MS, United States of America

## ABSTRACT

Many subclades within the large North American freshwater fish genus *Etheostoma* (Percidae) show brilliant male nuptial coloration during the spring spawning season. Traditionally, perceived differences in color were often used to diagnose closely related species. More recently, perceived differences in male nuptial color have prompted further investigation of potential biodiversity using genetic tools. However, cryptic diversity among *Etheostoma* darters renders male nuptial color as unreliable for detecting and describing diversity, which is foundational for research and conservation efforts of this group of stream fishes. *Etheostoma raneyi* (Yazoo Darter) is an imperiled, range-limited fish endemic to north-central Mississippi. Existing genetic evidence indicates cryptic diversity between disjunctly distributed *E. raneyi* from the Little Tallahatchie and Yocona river watersheds despite no obvious differences in male color between the two drainages. Analysis of morphological truss and geometric measurements and meristic and male color characters yielded quantitative differences in *E. raneyi* from the two drainages consistent with genetic evidence. Morphological divergence is best explained by differences in stream gradients between the two drainages. *Etheostoma faulkneri*, the Yoknapatawpha Darter, is described as a species under the unified species concept. The discovery of cryptic diversity within *E. raneyi* would likely not have occurred without genetic tools. Cryptic diversity among *Etheostoma* darters and other stream fishes is common, but an overreliance on traditional methods of species delimitation (e.g., identification of a readily observable physical character to diagnose a species) impedes a full accounting of the diversity in freshwater fishes in the southeastern United States.

Corresponding author
Ken A. Sterling,
kenneth.a.sterling@usda.gov

## INTRODUCTION

*Etheostoma raneyi* Suttkus and Bart (Yazoo Darter) is most closely related to other snubnose darters in western Tennessee and Kentucky, and Alabama (unranked clade name Adonia, *sensu* (*Near et al., 2011*; *Kozal et al., 2017*). The species is distributed across small tributaries of the Little Tallahatchie (L.T.R.) and Yocona rivers (Y.R.) of north-central Mississippi in the upper Yazoo River basin (Fig. 1). *Etheostoma raneyi* avoids the bottomland streams of the Mississippi Alluvial Plain and is limited to relatively higher-gradient, perennial streams

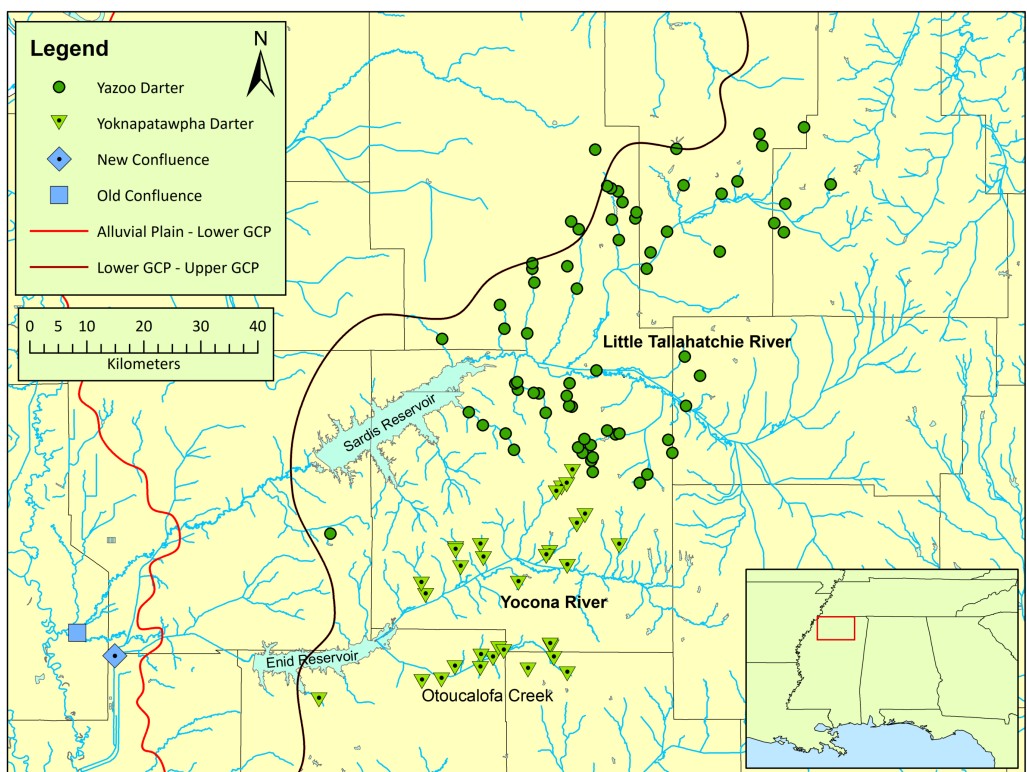

**Figure 1** **Map of the distribution of *Etheostoma raneyi* (Yazoo Darter) and *E. faulkneri* (Yoknapataw-pha Darter) in northern Mississippi.** Distribution is indicated by green circles in the Little Tallahatchie River drainage (*E. raneyi*) and green triangles in the Yocona River drainage (*E. faulkneri*). The location of the old confluence (blue square) of the two drainages is shown as well as the new confluence (blue diamond) after stream alterations; the approximate boundaries of the Mississippi Alluvial Plain, Lower Gulf Coastal Plain, and Upper Gulf Coastal Plain are indicated from west to east, respectively.

draining sandy geologic formations of the Upper Gulf Coastal Plain (*Stephenson, Logan & Waring, 1928*; *Randolph & Kennedy, 1974*; *Thompson & Muncy, 1986*; *Suttkus, Bailey & Bart Jr, 1994*; *Thompson, 2011*; *Sterling, Warren Jr & Henderson, 2013*).

A phylogenetic study of Coastal Plain snubnose darters in western Tennessee and Kentucky, and northern Mississippi, including *E. raneyi*, uncovered substantial genetic structure within and among species that was attributed to watershed configurations and the location of stream confluences between large drainages within the unfavorable lowland habitat of the Lower Gulf Coastal Plain and Mississippi Alluvial Plain (*Powers & Warren Jr, 2009*; *Keck & Etnier, 2005*) (Fig. 1). *Etheostoma raneyi* from the L.T.R. and Y.R. drainages were recovered as reciprocally monophyletic lineages, indicating that individuals from each drainage were evolutionarily divergent and distinct (*Powers & Warren Jr, 2009*). A more recent phylogenetic analysis using two genes and greater number of samples from across the range of the species also indicated that *E. raneyi* in the two drainages were independently evolving metapopulations (*Sterling et al., 2020*). Estimated time since divergence (0.4–0.8 my) was similar to estimates for snubnose darters in western Tennessee and Kentucky (*Kozal et al., 2017*).

The original description of *E. raneyi* did not indicate any geographic variation in appearance, male nuptial color, morphology, or meristics except for modal lateral-line scale counts between the L.T.R. and Y.R. (*Suttkus, Bailey & Bart Jr, 1994*). However, an examination of standard lengths (SL) between drainages showed that males and females from the Y.R. were longer compared with males and females from the L.T.R. (*Sterling, Warren Jr & Henderson, 2013*). Because the available evidence suggested differences in meristic and morphological characters and because the genetic evidence indicated that *E. raneyi* in the Y.R. were distinct, we investigated possible differences in male nuptial color, meristic characters, and morphology. The description of the new species presented here is based on published data and our new analyses.

## MATERIAL AND METHODS

This study was conducted with the approval of the University of Mississippi IACUC Committee (protocol 09-027), using annual collection permits issued to us from the Mississippi Museum of Natural Science (Mississippi Department of Wildlife, Fisheries, and Parks) for the years 2009–2014 and 2017–2018 (permit numbers: 0604091, 0513101, 0624112, 0622122, 0602132, 0610142, 0715163, 1010173).

### Male nuptial color

We examined male nuptial color by taking photographs of mature, live fish in the field. Males were captured by seine and deposited into a water-filled, opaque black bucket with a lid and supplemental oxygen (a bubbler) to prevent stress and subsequent loss of color. We used a small, clear Plexiglas photarium with a white foam squeeze plate to obtain images of a lateral view of the entire individual. We made all photographs using an Olympus Stylus TG-3 camera. Though one of us (K.S.) edited images for brightness and contrast, no alterations were made to hue or saturation. We made collections from February to April 2017 and March to April 2018 when males are at the peak of nuptial color (Table 1). We made a total of seven collections from six streams in the L.T.R. drainage ($n = 46$) and eight collections from five streams in the Y.R. drainage ($n = 33$). With the exception of male type specimens collected in 2018, all fishes were released after photographing them (see details under the Taxonomy subheading). We used resulting images to characterize colors and pigment patterns.

To assess possible color and pigment pattern differences between drainages following preliminary comparisons by one of us (K.S.), we asked three colleagues to score images of darters (see Table S1) for three characters: (1) the presence and density of black pigment in the pelvic and median fins, breast, branchiostegal membranes, chin, opercle, and cheek, which were scored as 1 = little pigment, 2 = moderate pigment, 3 = heavy pigment; (2) the presence and density of blue pigment in the same areas as for black pigment, which were scored identically; (3) presence of a clear window in the anal fin scored as 1 = window is small to non-existent and covers ≤3 membranes, blue pigment mostly extends to the belly, 2 = window is large and covers ≥3 membranes, blue pigment mostly does not extend to the belly.

**Table 1  Specimen data used for male nuptial color comparisons.** Stream name, date of sample, water temperature, and the number of males photographed are indicated.

| Drainage | Stream | Date | Temp. (°C) | n | Coordinates |
|---|---|---|---|---|---|
| L.T.R. | Bay Springs Branch | 2/23/2017 | 15 | 12 | 34.428, −89.395 |
| L.T.R. | Hurricane Creek | 3/16/2017 | 9.5 | 7 | 34.425, −89.496 |
| L.T.R. | Yellow Rabbit Creek | 4/7/2017 | 14 | 7 | 34.819, −89.106 |
| L.T.R. | Big Spring Creek | 4/17/2017 | 16 | 9 | 34.664, −89.413 |
| L.T.R. | Chewalla Creek | 3/22/2018 | 8.5 | 7 | 34.725, −89.305 |
| L.T.R. | Chewalla Creek | 3/22/2018 | 13 | 5 | 34.76, −89.333 |
| L.T.R. | Graham Mill Creek | 3/22/2018 | 13 | 4 | 34.503, −89.491 |
| | **Total** | | | 51 | |
| Y.R. | Morris Creek | 2/24/2017 | 15 | 12 | 34.282, −89.544 |
| Y.R. | Morris Creek | 3/8/2018 | 11.5 | 5 | 34.282, −89.544 |
| Y.R. | Morris Creek | 4/1/2018 | 15.5 | 3 | 34.282, −89.544 |
| Y.R. | Johnson Creek | 4/6/2017 | 14 | 5 | 34.124, −89.641 |
| Y.R. | Gordon Branch | 4/6/2017 | 13 | 1 | 34.14, −89.549 |
| Y.R. | Mill Creek | 1/1/2017 | – | 3 | 34.167, −89.52 |
| Y.R. | Mill Creek | 3/8/2018 | 9.5 | 5 | 34.167, −89.52 |
| Y.R. | U.T. Otoucalofa Creek | 3/23/2018 | 11 | 2 | 34.126, −89.611 |
| | **Total** | | | 36 | |

**Notes.**

L.T.R., Little Tallahatchie River (*Etheostoma raneyi*); Y.R., Yocona River (*E. faulkneri*); U.T., unnamed tributary.

Images were presented in random order and with no information on the location from which they were sampled. We calculated means and 95% confidence intervals of scores for each drainage from each of our colleagues results. Lastly, we also calculated the proportion of males having orange pigment in the anal fin from each drainage.

## Meristic, truss morphometric, and geometric morphometric analyses

We counted meristic characters following the methods of *Hubbs & Lagler (2004)* except we counted transverse scales from the origin of the anal fin diagonally toward the base of the spinous dorsal fin (*Page, 1983*). For comparison, we compiled meristic data for closely related Adonia snubnose darters (*Near et al., 2011*) from several sources in the literature (*Bailey & Etnier, 1988*; *Powers & Mayden, 2003*; *Kozal et al., 2017*). Counting methods were clearly different for one source (*Kozal et al., 2017*) compared with our methods and other data sources as evidenced by differences in modal counts that were consistently either one count higher or lower. In one instance (second dorsal fin rays) this difference in method was cited in the text. We adjusted counts for consistency with other sources of data, except for caudal peduncle scales because of small sample size and no clear modal count.

We made morphometric truss measurements following *Hubbs & Lagler (2004)* and *Humphries et al. (1981)* (Fig. 2). We measured distances between 28 pairs of points (digital calipers, nearest 0.1 mm) and converted them to thousandths of SL to remove the effects of differences in length (*Grabowski, Pease & Groeschel-Taylor, 2018*). We assessed distributions of each variable in PC-Ord ver. 6.21 (*McCune & Mefford, 2011*) and, as expected, variables had little skew and roughly normal distributions. We then estimated

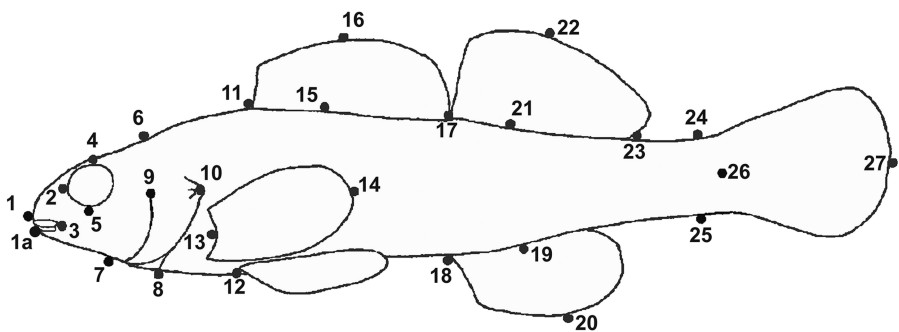

**Figure 2** Location of landmarks used for truss measurements (after *Powers & Mayden, 2003*). Numbers correspond to Table 10.

means and 95% confidence intervals for each variable using resampling with replacement and 10,000 samples (*Statistics.com LLC, 2009*). Specimens were primarily from our own collections now deposited with the National Museum of Natural History (Washington, D.C.) (USNM) or the Mississippi Museum of Natural Sciences (Jackson, MS) (MMNS). We obtained additional specimens from Tulane University Biodiversity Research Institute (Belle Chasse, LA) (TU). *Etheostoma raneyi* matures at about 28–30 mm SL (*Suttkus, Bailey & Bart Jr, 1994*), so we only used specimens >30 mm SL.

We used photographs of individual fish to perform geometric morphometric analyses (GMA) (*Klingenberg, 2011*) to test for and describe differences in shape between *E. raneyi* in the L.T.R. and Y.R. We chose to use this approach rather than more traditional multivariate analysis of linear truss measurements (e.g., *Suttkus, Bailey & Bart Jr, 1994*) because GMA describes changes in shape among all landmark points simultaneously and produces clear graphics of shape changes and effect sizes (e.g., misclassification tables) among groups (*Bookstein, 1991*; *Parsons, Robinson & Hrbek, 2003*; *Klingenberg, 2013*).

GMA requires images of each specimen, and a set of uniform landmark points plotted on each image before analysis. We photographed the lateral view of the left side of the entire fish using an Olympus Stylus TG-3 camera mounted on a vertical camera stand. A glass plate was placed on top of the fish to reduce curvature. Each image included a specimen ID label and a scale (mm). We imported images into tpsUtil (ver. 1.74, *Rohlf, 2017a*) and tpsDig (ver. 2.30, *Rohlf, 2017b*) software. We plotted, scaled, and digitized landmarks to produce Cartesian grid coordinates for each individual using 19 homologous landmarks (Fig. 3). Because preserved specimens are often vertically curved, we used tpsUtil (ver. 1.74, *Rohlf, 2017a*) to straighten landmark coordinates along the midline of the body.

We imported output from the tps software into MorphoJ software for all subsequent analyses (*Klingenberg, 2011*; *Klingenberg, 2018*). We compared shapes for males and females between drainages and compared shapes between sexes within each drainage (L.T.R. and Y.R.) (Table 2).

We used the outlier function in MorphoJ to remove individuals that may have biased the results and performed a least-squares full procrustes superimposition to remove bias resulting from differences in position and orientation among individuals (*Mitteroecker &*
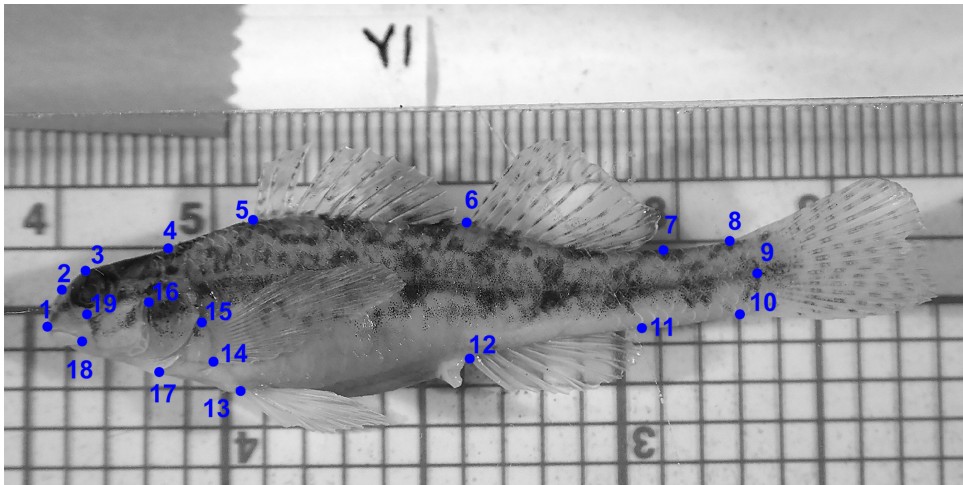

**Figure 3** **Landmark locations used for geometric morphological analyses (GMA).** The label Y1 in the photograph indicates individual number one from the Yocona River drainage.

*Gunz, 2009*; *Klingenberg, 2011*; *Klingenberg & Marugán-Lobón, 2013*). Pooled within-group multivariate regression of shape data (Procrustes coordinates) against $\log_{10}$ centroid size removed possible bias resulting from ontogenetic allometry; we used the resulting residual shape variation for all subsequent analyses in MorphoJ (*Loy et al., 1998*; *Klingenberg, 2011*). We used discriminant analysis (DA) to test for differences in shape for all comparisons. We estimated mean Procrustes distance (PD) and Mahalanobis distance (M) for each comparison and ran a permutation test (10,000 permutations) to estimate *p*-values. We used cross-validated classification tables to assess the reliability of the DA results and produced shape change graphics from DA output for all comparisons.

Preliminary results indicated shape changes between drainages that suggested possible differences in stream gradients. Therefore, we calculated watershed area ($km^2$) and stream gradient (relief/length of stream) for all streams with Yazoo Darters in each drainage using DeLorme Topo USA ver. 7.1.0 (*2007*) and estimated means. We used resampling with replacement to calculate 95% confidence intervals of mean values using Resampling Stats ver. 4.0 (*Statistics.com LLC, 2009*).

We evaluated all data and results from analyses under the unified species concept which recognizes that all species concepts attempt to define independently evolving metapopulations but use different methods for doing so (*de Queiroz, 2007*). The concept provides a useful framework for species delimitation by allowing evaluation of alternate hypotheses using all available evidence and recognizing that the emergence of diagnostic characters (e.g., genetic, morphological, or behavioral) among independently evolving lineages may or may not occur, thus accommodating cryptic diversity (see *Love Stowell et al., 2018*).

Data used for GMA (MorphoJ) are available from the open access MorphoBank (*O'Leary & Kaufman, 2012*) data repository at 10.7934/P3712. Fishes examined for the truss morphological measurements were borrowed from the Tulane University Biodiversity

**Table 2 Specimen data used for geometric morphological analyses (MorphoJ).** Locations and sample sizes for collections by drainage are indicated.

| Drainage | Stream | Coordinates | Male n | Female n |
|---|---|---|---|---|
| L.T.R. | Yellow Rabbit Creek | 34.819, −89.106 | 6 | 5 |
| L.T.R. | Chewalla Creek | 34.726, −89.305 | 13 | 7 |
| L.T.R. | Bay Springs Branch | 34.428, −89.395 | 8 | 0 |
| L.T.R. | Big Spring Creek | 34.664, −89.413 | 12 | 9 |
| L.T.R. | Shelby Creek | 34.844, −89.039 | 1 | 2 |
| L.T.R. | Fice Creek | 34.421, −89.247 | 1 | 1 |
| L.T.R. | U.T. Tippah River | 34.682, −89.281 | 1 | 1 |
| L.T.R. | U.T. Tippah River | 34.712, −89.254 | 0 | 4 |
| L.T.R. | Oak Chewalla | 34.583, −89.511 | 0 | 2 |
| L.T.R. | Chilli Creek | 34.682, −89.173 | 0 | 1 |
| L.T.R. | Mitchell Creek | 34.521, −89.203 | 0 | 1 |
| L.T.R. | Puskus Creek | 34.447, −89.345 | 1 | 5 |
| L.T.R. | Cypress Creek | 34.383, −89.299 | 5 | 6 |
| L.T.R. | Lee Creek | 34.498, −89.457 | 0 | 1 |
| L.T.R. | Wagner Creek | 34.768, −89.229 | 2 | 2 |
| L.T.R. | Hurricane Creek | 34.425, −89.496 | 2 | 6 |
|  | **Total** |  | **52** | **53** |
| Y.R. | Johnson Creek | 34.124, −89.641 | 0 | 1 |
| Y.R. | Morris Creek | 34.282, −89.544 | 4 | 9 |
| Y.R. | Mill Creek | 34.167, −89.52 | 5 | 1 |
| Y.R. | Gordon Branch | 34.14, −89.549 | 9 | 4 |
| Y.R. | Yellow Leaf Creek | 34.376, −89.421 | 2 | 5 |
| Y.R. | Pumpkin Creek | 34.285, −89.445 | 14 | 9 |
| Y.R. | U.T. Otoucalofa Creek | 34.126, −89.611 | 6 | 19 |
| Y.R. | Taylor Creek | 34.297, −89.589 | 3 | 6 |
| Y.R. | Smith Creek | 34.168, −89.439 | 1 | 0 |
| Y.R. | Splinter Creek | 34.236, −89.635 | 5 | 6 |
|  | **Total** |  | **49** | **60** |

Notes.

L.T.R., Little Tallahatchie River (*Etheostoma raneyi*); Y.R., Yocona River (*E. faulkneri*); U.T., unnamed tributary.

Research Institute (Belle Chasse, LA) (TU) or were from our own collections now archived at the National Museum of Natural History (Washington, D.C.) (USNM) and the Mississippi Museum of Natural Sciences (Jackson, MS) (MMNS). Images used for male nuptial color comparisons are available in the Supplemental Information.

The electronic version of this article in Portable Document Format (PDF) will represent a published work according to the International Commission on Zoological Nomenclature (ICZN), and hence the new name contained in the electronic version are effectively published under that Code from the electronic edition alone. This published work and the nomenclatural acts it contains have been registered in ZooBank, the online registration system for the ICZN. The ZooBank LSIDs (Life Science Identifiers) can be resolved and the associated information viewed through any standard web browser by

appending the LSID to the prefix http://zoobank.org/. The LSID for this publication
is: urn:lsid:zoobank.org:pub:6C5BEC69-22A1-4008-84D3-CA62719AE152. The online
version of this work is archived and available from the following digital repositories: PeerJ,
PubMed Central and CLOCKSS.

# RESULTS

## Male nuptial color

Preliminary examination of photographed male darters indicated that possible color or
pigment pattern differences were subtle and appeared limited to: (1) generally more
extensive and denser black pigment in the pectoral and median fins and the breast, cheek,
opercle, and ventral portion of the head region for individuals sampled in the L.T.R.; (2)
generally more extensive and denser blue pigment in the same areas for individuals from
the Y.R.; (3) complete absence or a small clear window in the anal fin for fish from the Y.R.
compared with windows always being present and larger in the L.T.R.; (4) orange pigment
occasionally present in the anal fin of fish from the L.T.R. and never present in fish from
the Y.R. (Fig. 4).

Independent scoring of these features revealed that these average differences are
consistent between the drainages (Fig. 5). Blue pigment was more dense and extensive
in the Y.R. males, but black pigment was generally more dense and extensive in the L.T.R.
males. Clear windows in the anal fin were generally smaller or absent in the Y.R. individuals
(Fig. 4). Orange pigment in the anal fin was not found in fish from the Y.R., but was present
in 17.7% of 46 males from four of six streams sampled in the L.T.R.

## Meristics and truss measurements

Count data for lateral-line scales show a bimodal distribution that differ by two scales
between individuals from the Y.R. and L.T.R. (Table 3), which is consistent with the counts
of *Suttkus, Bailey & Bart Jr (1994)*. There are no other modal differences between the Y.R.
and the L.T.R. drainages (Tables 4–9).

Mean proportional values for truss measurements indicate that individuals from the
Y.R. are more robust with shorter heads and snouts relative to individuals from the L.T.R.
Dorsal fins are also longer in the Y.R., but gape width and the position of the mouth relative
to the tip of the snout shows no difference. Females in the L.T.R. have a wider inter-orbital
width, but females in the Y.R. had a wider body at the pectoral fin insertion. Overall, 13
of 29 characters did not have overlapping 95% CIs for at least one sex between drainages.
Mean SL for both males and females in the Y.R. is longer than for males and females in the
L.T.R. (Table 10).

## GMA

The outlier function in MorphoJ identified two females and two males in the L.T.R. as
potentially biasing results and these were removed. No individuals from the Y.R. were
identified as possible outliers.

There were significant differences in shape between females in the L.T.R. ($n = 51$) and
Y.R. ($n = 60$) drainages (DA, PD $= 0.014$, $p < 0.0001$; $M = 3.59$, $p < 0.0001$). Classification

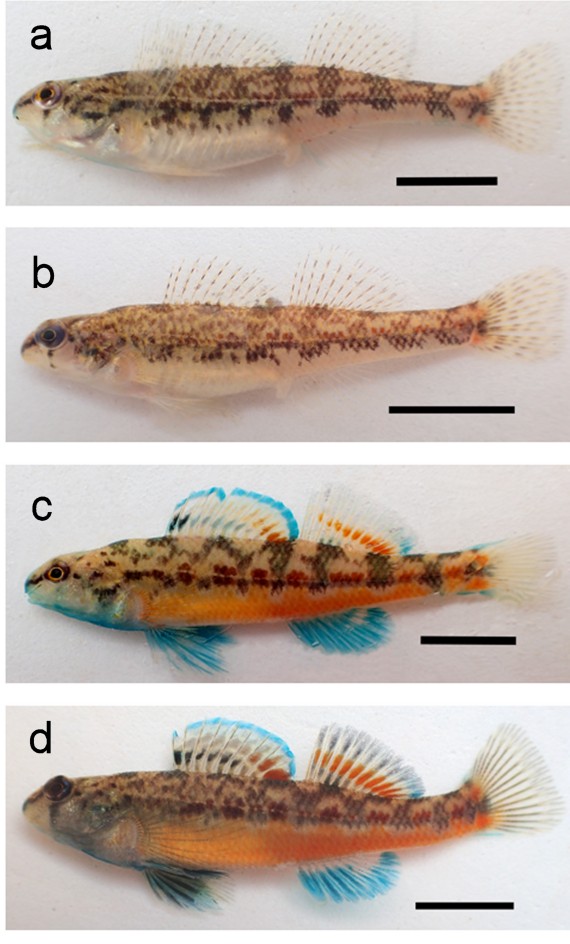

**Figure 4** **Photographs of *Etheostoma faulkneri* (Yoknapatawpha Darter) and *E. raneyi* (Yazoo Darter).** Letters correspond to *E. faulkneri* female allotype (A), female *E. raneyi* (B), *E. faulkneri* male holotype (C), and male *E. raneyi* (D); scale bar = 1 cm.

of specimens to groups shows that 12% of individuals from the L.T.R. were incorrectly assigned to the Y.R. drainage and 17% of individuals from the Y.R. were incorrectly assigned to the L.T.R. drainage. Yocona River females had shorter snouts and heads (points 4 and 16), greater body depth, longer second dorsal fin (point 7), longer anal fin associated with an anterior shift in the insertion (point 12), and the posterior edge of the hypural plate shows a shift anteriorly (Fig. 6).

There were significant differences in shape between males in the L.T.R. ($n = 50$) and Y.R. ($n = 49$) (DA, PD = 0.009, $p < 0.0001$; $M = 2.78$, $p < 0.0001$). Classification of specimens to groups shows that 32% of individuals from the L.T.R. were incorrectly assigned to the Y.R. and 29% of individuals from the Y.R. were incorrectly assigned to the L.T.R. Shape changes between drainages show that Y.R. males had shorter snouts and heads (points 4 and 16), greater body depth, and longer second dorsal fins (point 7) (Fig. 6).

Sterling and Warren, Jr (2020), *PeerJ*, DOI 10.7717/peerj.9807

**Table 3** **Compilation of frequency distributions of lateral line scale counts for six species of snubnose darters and one undescribed form.** Modal counts are in bold; sources of data not cited in the table are: *Etheostoma cervus* (*Bailey & Etnier, 1988*; *Powers & Mayden, 2003*; *Kozal et al., 2017*); *E. pyrrhogaster, E. zonistium, E.* cf. *zonistium*, and *E. cyanoprosopum* (*Bailey & Etnier, 1988*; *Kozal et al., 2017*).

| | 36 | 37 | 38 | 39 | 40 | 41 | 42 | 43 | 44 | 45 | 46 | 47 | 48 | 49 | 50 | 51 | 52 | 53 | n | Mean | S.D. |
|---|---|---|---|---|---|---|---|---|---|---|---|---|---|---|---|---|---|---|---|---|---|
| *E. cervus* | 2 | 11 | 30 | **49** | 40 | 29 | 8 | 3 | | 2 | | | | | | | | | 174 | 39.5 | 1.51 |
| *E. pyrrhogaster* | | | 8 | 10 | 40 | 44 | **47** | 26 | 13 | 10 | 3 | | | | | | | | 201 | 42.5 | 1.73 |
| *E. zonistium* (Tennessee R.) | | | | 5 | 7 | 18 | 38 | 45 | 61 | **85** | 65 | 50 | 50 | | | | | | 424 | 45.8 | 2.13 |
| *E.* cf. *zonistium* (Hatchie R.) | | | | | | 5 | 5 | 6 | 12 | **33** | 22 | 15 | 15 | 5 | 2 | 1 | 1 | | 122 | 45.7 | 2.11 |
| *E. cyanoprosopum* | | | | | | | | | 1 | 2 | 11 | 18 | **31** | 23 | 15 | 11 | 3 | 2 | 117 | 48.5 | 1.73 |
| *E. raneyi* (*Suttkus, Bailey & Bart Jr, 1994*) | | | | | | 1 | 7 | 15 | 24 | **34** | 25 | 32 | 11 | 11 | 1 | 2 | 1 | 1 | 165 | 45.7 | 2.09 |
| *E. faulkneri* (*Kozal et al., 2017*) | | | | | | | | | 5 | **10** | 7 | 5 | 4 | 4 | 1 | | | | 36 | 46.3 | 1.70 |
| *E. raneyi* (new data) | | | | | | | 3 | 3 | 9 | 19 | 21 | **34** | 21 | 19 | 15 | 7 | 1 | | 152 | 47.2 | 2.10 |
| *E. faulkneri* (new data) | | | | | | 1 | 5 | 6 | 16 | **29** | 22 | 19 | 13 | 7 | 1 | 1 | | | 120 | 45.8 | 1.88 |

**Notes.**

S.D., standard deviation.
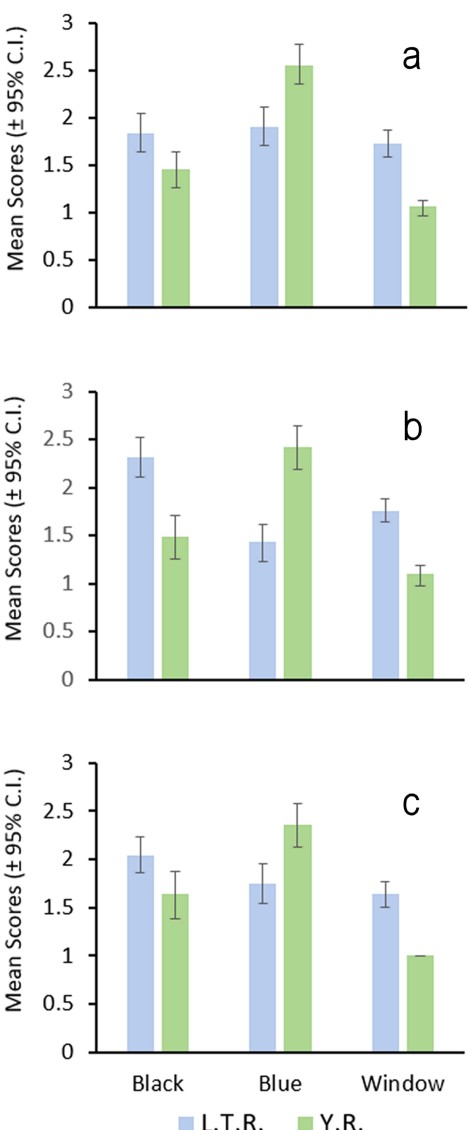

**Figure 5  Bar graph of the results of male nuptial color surveys.** Each graph (A, B, C) represents mean values (±95% CI) scored by one of three colleagues for each character (Black, Blue, Window, see text for definitions of characters); L.T.R. = Little Tallahatchie River, $n = 51$; Y.R. = Yocona River, $n = 36$.

There were significant differences in shape between males ($n = 50$) and females ($n = 51$) within the L.T.R. drainage (DA, PD = 0.015, $p < 0.0001$; M = 3.05, $p < 0.0001$). Classification of specimens to groups shows that 20% of females were incorrectly classified as males and 25% of males were incorrectly classified as females. Relative shape changes between males and females show that females are thinner with longer caudal peduncles, the insertion of the pectoral and pelvic fins shifted anteriorly, and the insertion of the anal fin shifted posteriorly relative to males (Fig. 7).

There were also significant differences in shape between males ($n = 49$) and females ($n = 60$) within the Y.R. drainage (DA, PD = 0.013, $p < 0.0001$; M = 4.63, $p < 0.0001$).

**Table 4** **Compilation of frequency distributions of transverse scale counts for six species of snubnose darters and one undescribed form.** Modal counts are in bold; sources of data not cited in the table are: *Etheostoma cervus* (*Powers & Mayden, 2003*; *Kozal et al., 2017*); *E. pyrrhogaster*, *E. zonistium*, *E. cf. zonistium*, and *E. cyanoprosopum* (*Kozal et al., 2017*).

| | 10 | 11 | 12 | 13 | 14 | 15 | 16 | 17 | 18 | n | Mean | S.D. |
|---|---|---|---|---|---|---|---|---|---|---|---|---|
| *E. cervus* | 3 | 20 | **32** | 11 | 3 | | | | | 69 | 11.9 | 0.89 |
| *E. pyrrhogaster* | 1 | **16** | 8 | 1 | 1 | | | | | 27 | 11.4 | 0.80 |
| *E. zonistium* (Tennessee R.) | | | 27 | **82** | 63 | 9 | | | | 181 | 13.3 | 0.78 |
| *E. cf. zonistium* (Hatchie R.) | 4 | **47** | 7 | 1 | | | | | | 59 | 11.1 | 0.50 |
| *E. cyanoprosopum* | | | 3 | 13 | 21 | **35** | 6 | 1 | 3 | 82 | 14.5 | 1.21 |
| *E. raneyi* (*Suttkus, Bailey & Bart Jr, 1994*) | | 2 | 36 | **57** | 51 | 16 | 3 | | | 165 | 13.3 | 1.02 |
| *E. faulkneri* (*Kozal et al., 2017*) | | 9 | 10 | **17** | | | | | | 36 | 13.7 | 5.34 |
| *E. raneyi* (new data) | | 17 | 42 | **75** | 19 | | | | | 153 | 12.6 | 0.84 |
| *E. faulkneri* (new data) | | 2 | 38 | **61** | 19 | 1 | | | | 121 | 12.8 | 0.74 |

**Notes.**
S.D., standard deviation.

**Table 5** **Compilation of frequency distributions of caudal peduncle scale counts for six species of snubnose darters and one undescribed form.** Modal counts are in bold; sources of data not cited in the table are: *Etheostoma cervus* (*Bailey & Etnier, 1988*; *Powers & Mayden, 2003*; *Kozal et al., 2017*); *E. pyrrhogaster*, *E. zonistium*, *E. cf. zonistium*, and *E. cyanoprosopum* (*Bailey & Etnier, 1988*; *Kozal et al., 2017*).

| | 14 | 15 | 16 | 17 | 18 | 19 | 20 | n | Mean | S.D. |
|---|---|---|---|---|---|---|---|---|---|---|
| *E. cervus* | 2 | 9 | 35 | **69** | 17 | 1 | 1 | 134 | 16.7 | 0.90 |
| *E. pyrrhogaster* | 4 | 18 | **19** | 14 | 1 | | | 56 | 15.8 | 0.96 |
| *E. zonistium* (Tennessee R.) | | 6 | 38 | **148** | 68 | 32 | 2 | 294 | 17.3 | 0.92 |
| *E. cf. zonistium* (Hatchie R.) | 2 | 18 | **41** | 36 | | | | 97 | 16.1 | 0.79 |
| *E. cyanoprosopum* | | | 5 | **45** | 28 | 26 | 8 | 112 | 17.9 | 1.05 |
| *E. raneyi* (*Suttkus, Bailey & Bart Jr, 1994*) | | 16 | 57 | **75** | 13 | 4 | | 165 | 16.6 | 0.86 |
| *E. faulkneri* (*Kozal et al., 2017*) | | 9 | **15** | 10 | 1 | | | 35 | 16.1 | 0.82 |
| *E. raneyi* (new data) | 1 | 11 | 37 | **66** | 30 | 4 | | 149 | 16.8 | 0.94 |
| *E. faulkneri* (new data) | | | 9 | **59** | 29 | 11 | | 108 | 17.4 | 0.78 |

**Notes.**
S.D., standard deviation.

Cross-validated classification of specimens to groups shows that 3% of females were incorrectly classified as males and 5% of males were incorrectly classified as females. Relative shape changes between males and females show that females are thinner, the snout is shorter, the anal and second dorsal fins are shorter, the insertion of the pectoral and pelvic fins shifts anteriorly, and the insertion of the anal fin shifts posteriorly relative to males (Fig. 7).

Comparison of watershed area and stream gradients between drainages show that *E. raneyi* in the L.T.R. mostly occur in larger streams with lower gradients compared with the Y.R. (Fig. 8, Table S1). Mean watershed area is more than twice as large in the L.T.R., and mean gradient is about 45% greater in the Y.R. drainage. Confidence intervals do not overlap for area or gradient.

**Table 6 Compilation of frequency distributions of first dorsal fin spine counts for six species of snubnose darters and one undescribed form.** Modal counts are in bold; sources of data not cited in the table are: *Etheostoma cervus* (*Bailey & Etnier, 1988*; *Powers & Mayden, 2003*; *Kozal et al., 2017*); *E. pyrrhogaster*, *E. zonistium*, *E.* cf. *zonistium*, and *E. cyanoprosopum* (*Bailey & Etnier, 1988*; *Kozal et al., 2017*).

| | 9 | 10 | 11 | 12 | n | Mean | S.D. |
|---|---|---|---|---|---|---|---|
| *E. cervus* | | 59 | **76** | 1 | 136 | 10.6 | 0.51 |
| *E. pyrrhogaster* | 8 | **42** | 27 | 1 | 78 | 10.3 | 0.66 |
| *E. zonistium* (Tennessee R.) | 20 | **252** | 96 | 2 | 370 | 10.2 | 0.54 |
| *E.* cf. *zonistium* (Hatchie R.) | 8 | **99** | 15 | | 122 | 10.1 | 0.43 |
| *E. cyanoprosopum* | 4 | **60** | 48 | 2 | 114 | 10.4 | 0.59 |
| *E. raneyi* (*Suttkus, Bailey & Bart Jr, 1994*) | 14 | **114** | 35 | 2 | 165 | 10.2 | 0.57 |
| *E. faulkneri* (*Kozal et al., 2017*) | 2 | **31** | 3 | | 36 | 10.0 | 0.38 |
| *E. raneyi* (new data) | 18 | **111** | 24 | | 153 | 10.0 | 0.52 |
| *E. faulkneri* (new data) | 8 | **86** | 21 | | 115 | 10.1 | 0.49 |

**Notes.**
S.D., standard deviation.

**Table 7 Compilation of frequency distributions of second dorsal fin ray counts for six species of snubnose darters and one undescribed form.** Modal counts are in bold; sources of data not cited in the table are: *Etheostoma cervus* (*Bailey & Etnier, 1988*; *Powers & Mayden, 2003*; *Kozal et al., 2017*); *E. pyrrhogaster*, *E. zonistium*, *E.* cf. *zonistium*, and *E. cyanoprosopum* (*Bailey & Etnier, 1988*; *Kozal et al., 2017*).

| | 9 | 10 | 11 | 12 | 13 | n | Mean | S.D. |
|---|---|---|---|---|---|---|---|---|
| *E. cervus* | | | **53** | 40 | 4 | 97 | 11.5 | 0.58 |
| *E. pyrrhogaster* | | | **41** | 34 | 3 | 78 | 11.5 | 0.58 |
| *E. zonistium* (Tennessee R.) | 3 | 58 | **249** | 59 | | 369 | 11.0 | 0.59 |
| *E.* cf. *zonistium* (Hatchie R.) | 1 | 21 | **68** | 2 | | 92 | 10.8 | 0.49 |
| *E. cyanoprosopum* | 2 | 26 | **81** | 3 | | 112 | 10.8 | 0.52 |
| *E. raneyi* (*Suttkus, Bailey & Bart Jr, 1994*) | 1 | 50 | **105** | 9 | | 165 | 10.7 | 0.56 |
| *E. faulkneri* (*Kozal et al., 2017*) | | 5 | **28** | 3 | | 36 | 10.9 | 0.47 |
| *E. raneyi* (new data) | 3 | 57 | **86** | 6 | | 152 | 10.6 | 0.60 |
| *E. faulkneri* (new data) | 1 | 53 | **61** | | | 115 | 10.5 | 0.52 |

**Notes.**
S.D., standard deviation.

## Taxonomy and synonymy

*Etheostoma faulkneri* Sterling and Warren, Yoknapatawpha Darter (Fig. 4)

urn:lsid:zoobank.org:act:B9FE97A8-A86C-41A9-9CD6-2929362E0C22

*Etheostoma* (*Ulocentra*) sp. (*Bouchard, 1974*): 41 (distribution).

*Etheostoma* sp. *Clemmer & Suttkus, 1975*: 8 (listing of an undescribed species of darter in the upper Yazoo River basin, categorized as rare). *Jenkins, 1976*: 644 (listing of an undescribed species, distribution).

*Etheostoma* (*Ulocentra*) sp. *Kuehne & Barbour, 1983*: 99–100 (brief characterization of the Yazoo Darter and distribution). *Knight & Cooper, 1987*: 31–32 (brief description of watershed-scale habitat before additional extensive habitat alteration), 36 (brief

**Table 8 Compilation of frequency distributions of anal fin ray counts for six species of snubnose darters and one undescribed form.** Modal counts are in bold; sources of data not cited in the table are: *Etheostoma cervus* (*Bailey & Etnier, 1988*; *Powers & Mayden, 2003*; *Kozal et al., 2017*); *E. pyrrhogaster, E. zonistium, E. cf. zonistium,* and *E. cyanoprosopum* (*Bailey & Etnier, 1988*; *Kozal et al., 2017*).

|  | 5 | 6 | 7 | 8 | 9 | n | Mean | S.D. |
|---|---|---|---|---|---|---|---|---|
| *E. cervus* |  | 2 | **74** | 58 | 2 | 136 | 7.4 | 0.55 |
| *E. pyrrhogaster* |  |  | 18 | **52** | 8 | 78 | 7.9 | 0.57 |
| *E. zonistium* (Tennessee R.) | 2 | 30 | **260** | 54 | 1 | 347 | 7.1 | 0.52 |
| *E. cf. zonistium* (Hatchie R.) |  | 23 | **64** | 5 |  | 92 | 6.8 | 0.52 |
| *E. cyanoprosopum* |  | 10 | **70** | 34 |  | 114 | 7.2 | 0.59 |
| *E. raneyi* (*Suttkus, Bailey & Bart Jr, 1994*) |  | 11 | **133** | 21 |  | 165 | 7.1 | 0.44 |
| *E. faulkneri* (*Kozal et al., 2017*) |  | 2 | **26** | 8 |  | 36 | 7.2 | 0.51 |
| *E. raneyi* (new data) |  | 11 | **99** | 9 |  | 119 | 7.0 | 0.41 |
| *E. faulkneri* (new data) |  | 7 | **86** | 9 |  | 102 | 7.0 | 0.40 |

**Notes.**
S.D., standard deviation.

**Table 9 Compilation of frequency distributions of pectoral fin ray counts for six species of snubnose darters and one undescribed form.** Modal counts are in bold; sources of data not cited in the table are: *Etheostoma cervus* (*Bailey & Etnier, 1988*; *Powers & Mayden, 2003*; *Kozal et al., 2017*); *E. pyrrhogaster, E. zonistium, E. cf. zonistium,* and *E. cyanoprosopum* (*Bailey & Etnier, 1988*; *Kozal et al., 2017*).

|  | 12 | 13 | 14 | 15 | 16 | n | Mean | S.D. |
|---|---|---|---|---|---|---|---|---|
| *E. cervus* |  | 20 | **72** | 5 |  | 97 | 13.8 | 0.49 |
| *E. pyrrhogaster* |  | 27 | **41** |  |  | 68 | 13.6 | 0.49 |
| *E. zonistium* (Tennessee R.) | 1 | 98 | **328** | 63 |  | 490 | 13.9 | 0.58 |
| *E. cf. zonistium* (Hatchie R.) |  | 38 | **83** | 1 |  | 122 | 13.7 | 0.48 |
| *E. cyanoprosopum* | 1 | 8 | **87** | 18 |  | 114 | 14.1 | 0.50 |
| *E. raneyi* (*Suttkus, Bailey & Bart Jr, 1994*) |  | 11 | **126** | 28 |  | 165 | 14.1 | 0.48 |
| *E. faulkneri* (*Kozal et al., 2017*) |  | 3 | **28** | 4 | 1 | 36 | 14.1 | 0.55 |
| *E. raneyi* (new data) |  | 11 | **86** | 15 |  | 112 | 14.0 | 0.48 |
| *E. faulkneri* (new data) |  | 10 | **80** | 10 |  | 100 | 14.0 | 0.45 |

**Notes.**
S.D., standard deviation.

description of meso-habitat and distribution within the Otoucalofa Creek drainage), Table 1 (occurrence record).

*Etheostoma* sp. *Page & Burr, 1991*: 302, pl. 43, map 345 (brief characterization of the Yazoo Darter and distribution.

*Etheostoma raneyi*, *Suttkus, Bailey & Bart Jr, 1994*: 98–109 (distribution, habitat, description of male and female color, pigment patterns, meristic characters, and morphological measurements subsumed under the description of *Etheostoma raneyi*), fig. 8 (ordination of morphometric data), fig. 9 (distribution). *Johnston & Haag, 1996*: 47–60 (life history, distribution). *Ross & Slack, 2000*: 1 (conservation status, distribution), fig. 1 (distribution), fig. 4 (photograph of a male Yazoo Darter from an unknown location). (*Ross, 2001*): 483–484 (distribution and general description and life history account,

**Table 10  Means and 95% confidence intervals (CI) for measurements of morphometric truss variables in thousandths of SL (except for SL) for *Etheostoma faulkneri* and *E. raneyi* males and females.** Variable labels correspond to landmarks in Fig. 2; measurements in bold font with asterisks indicate that 95% confidence intervals do not overlap within sexes between drainages.

| Variable | *E. faulkneri* males (*n* = 43) mean (95% CI) | *E. raneyi* males (*n* = 68) mean (95% CI) | *E. faulkneri* females (*n* = 51) mean (95% CI) | *E. raneyi* females (*n* = 77) mean (95% CI) |
|---|---|---|---|---|
| SL | *42.9 (41.3–44.6) | *39.7 (38.4–41.1) | *39.3 (38.4–40.1) | *37.3 (36.3–38.3) |
| 1–3 | *63 (62.6–64.9) | *67.6 (66.5–68.5) | *61.2 (60.3–62) | *66 (65.2–66.8) |
| 1a–3 | *56.5 (55.1–57.8) | *59.6 (58.6–60.6) | *54.2 (53.4–55.1) | *58.4 (57.5–59.4) |
| 1–1a | 7.3 (6.3–8.4) | 7.9 (7.3–8.6) | 6.9 (6.4–7.5) | 7.6 (7.1–8.1) |
| 3–3 | 73.1 (71.4–74.9) | 70.7 (69.3–72) | 66.8 (65.5–68) | 68.8 (67.8–69.8) |
| 1–2 | *47.4 (45.9–48.9) | *51.8 (50.3–53.2) | *45.6 (44–46.9) | *48.6 (47.4–49.7) |
| 3–5 | 36 (34.4–37.7) | 36 (34.5–37.7) | 30.6 (29.3–31.9) | 32.2 (30.7–33.8) |
| 1–9 | *159.2 (157–161.3) | *164.8 (162.8–166.9) | *156.9 (155.2–158.5) | *162.8 (160.8–164.7) |
| 9–10 | 88.2 (86.3–90.2) | 89.3 (87.8–90.7) | 84.4 (83–85.9) | 84.3 (83–85.6) |
| 4–4 | 44.1 (43.1–45.1) | 45.8 (44.9–46.8) | *42.1 (40.9–43.2) | *45.6 (44.9–46.4) |
| 13–13 | 141.9 (138.2–145.7) | 139.7 (137.5–141.8) | *145.4 (142.6–148.3) | *137.6 (135.3–139.8) |
| 13–14 | 272 (266.3–277) | 272.7 (269.5–275.9) | 271.7 (268–275.2) | 271.7 (269–274.5) |
| 15–16 | 137 (134–139.8) | 140.3 (138.1–142.6) | 126.7 (124.2–129.1) | 129.9 (120.5–133.5) |
| 19–20 | 118.7 (115–122.2) | 121.4 (118.3–124.7) | 118.5 (115.7–121.4) | 117.1 (114.5–119.7) |
| 21–22 | 155.1 (151.7–158.5) | 154.1 (151.6–156.5) | 152.4 (149.3–155.3) | 151.7 (149.6–153.7) |
| 23–24 | 217.1 (214–220.1) | 216.2 (211.7–220.8) | 215.5 (211.3–219.5) | 218.1 (214.9–220.9) |
| 24–25 | 104.2 (102.4–106) | 101.5 (100.3–102.7) | 94.7 (93.5–96) | 95.8 (94.9–96.7) |
| 26–27 | 222.4 (218.9–225.7) | 227.9 (224.2–231.9) | 227.6 (224.6–230.5) | 226.6 (224.3–229) |
| 1–6 | *196.3 (194.6–197.9) | *204.2 (202.4–206) | *195.7 (194.3–197.1) | *202.2 (200.6–203.8) |
| 1–7 | *106.6 (102.9–110.1) | *122.8 (120.4–125.2) | *110.4 (107.4–113.4) | *117.3 (115.2–119.3) |
| 6–11 | 143.7 (141.5–145.9) | 142.4 (140.6–144.3) | 145.3 (143–147.8) | 142.9 (141.3–144.5) |
| 7–12 | *183.4 (179.9–187) | *175.8 (172.4–179.2) | 176.5 (172.9–179.9) | 179 (175.7–182.4) |
| 11–12 | *212.7 (208.5–216.9) | *204.3 (201.5–207.1) | *214.8 (210.7–219) | *196.3 (193.7–198.8) |
| 11–17 | *298.7 (295–302.3) | *287.1 (283.1–291.2) | *286.4 (283.2–289.5) | *270.6 (266.9–274.4) |
| 11–18 | *344.4 (341.4–347.6) | *335.2 (332–338.4) | *343.5 (340.3–346.7) | *333.5 (330.6–336.5) |
| 12–17 | 371.5 (366.2–376.7) | 370.9 (367.3–374.5) | 377.1 (373–381.3) | 368.3 (364.5–372) |
| 17–18 | *187.2 (183.4–191.2) | *176.9 (174.6–179.3) | 171.3 (169–173.6) | 167.6 (165.8–169.4) |
| 17–25 | 396.3 (392.8–399.8) | 393.7 (390.4–397) | 389.7 (386.9–392.6) | 388.7 (386.4–391.1) |
| 18–24 | 409.6 (405.9–413.3) | 413.5 (410.5–416.5) | 401.6 (398–405.2) | 402.4 (399.6–405.2) |

conservation status). *Adams & Warren Jr, 2005*: fig. 4 (scatterplot showing post-drought CPUE and immigration probabilities in stream reaches that went dry during an extended drought), Appendix 1 (list of immigration probabilities and standard deviation). *Powers & Warren Jr, 2009* (phylogeography and distribution). *Near et al., 2011*: fig. 3C (phylogenetic relationships among darters), fig. 4 (phylogenetic classification among darters). *Page & Burr, 2011*: 558, pl. 49 (brief characterization of the Yazoo Darter and distribution). *Schaefer, Clark & Warren Jr, 2012*: Appendix (occurrence and abundance). *Sterling et al., 2012*: 859–872 (population genetics and effects of habitat alteration on gene flow). *Sterling, Warren Jr & Henderson, 2013*: 816–842 (distribution, abundance, life history, and conservation assessment). *Kozal et al., 2017*: Table 1 (collection data), Tables 4–10 (meristic data),

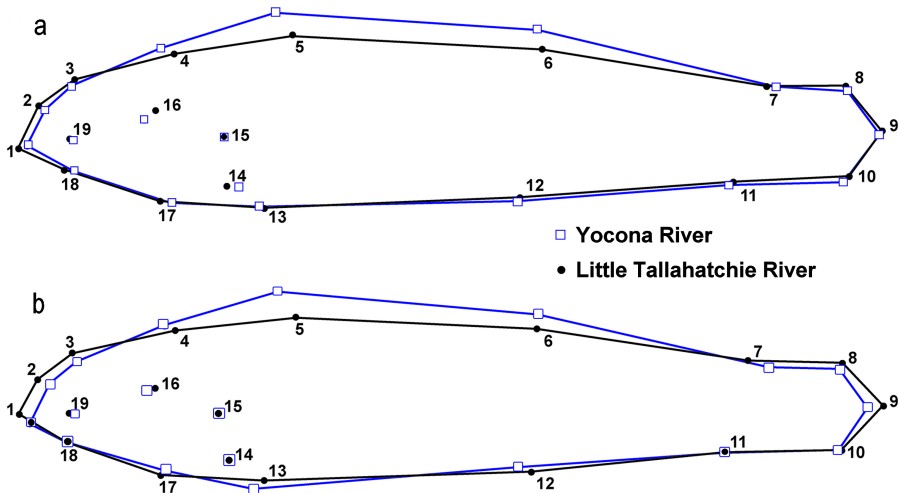

**Figure 6** Results from lateral view geometric morphological analyses (GMA, MorphoJ) indicating shape changes for male (A) and female (B) *Etheostoma raneyi* and *E. faulkneri* between drainages. Numbers represent fixed points in the analysis (see Fig. 3).

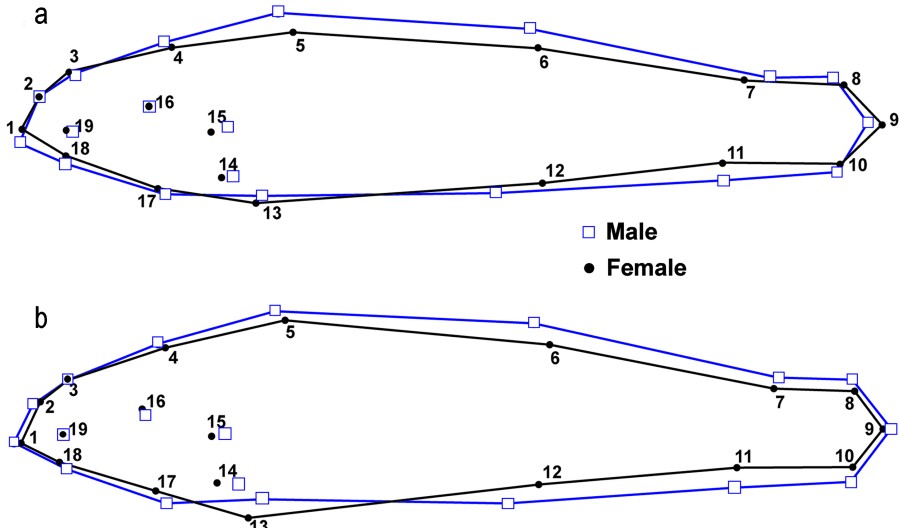

**Figure 7** Results from lateral view geometric morphological analyses (GMA) indicating shape changes for male and female *Etheostoma raneyi* (A) (Little Tallahatchie River) and *E. faulkneri* (B) (Yocona River) between sexes within drainages. Numbers represent fixed points in the analysis (see Fig. 3).

279, Table 11, and fig. 7 (results from meristic analyses), 279, fig. 5, (phylogenetic results). *Sterling & Warren Jr, 2017*: 1223–1235 (microhabitat use). *Sterling et al., 2020*: phylogenetic relationships.

*Holotype*: Adult male, USNM439004 (Fig. 4), 43.3 mm SL, Morris Creek, tributary to the Yocona River (Upper Yazoo River system) at County Road 321, 9.28 km south of the

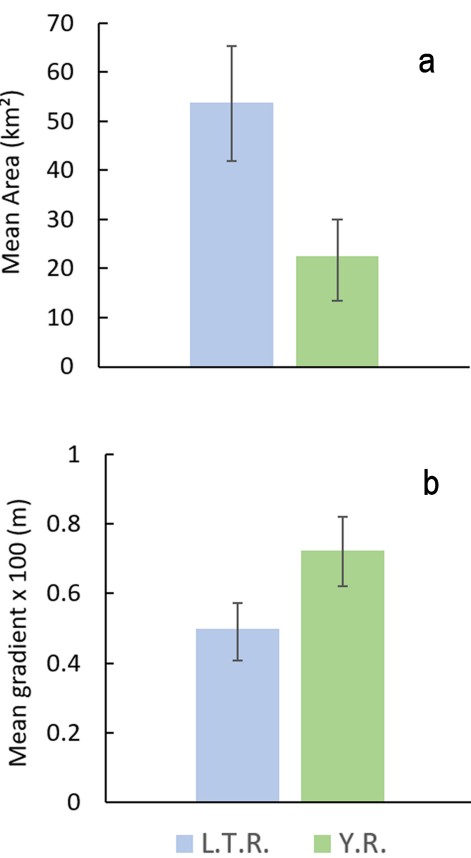

**Figure 8** **Mean watershed area (A) and stream gradient (B) (±95% CI) for all tributary streams with _Etheostoma raneyi_ and _E. faulkneri_.** Abbreviations indicate Y.R. = Yocona River and L.T.R. = Little Tallahatchie River drainages.

Lafayette County courthouse in Oxford (34.283, −89.544), Lafayette County, MS, 1 April 2018, B.D. Sterling, K.A. Sterling, and W.M. Sterling.

  _Allotype_: Adult female, USNM439005 (Fig. 4), 41.2 mm SL, collected with the holotype.

  _Paratopotypes_: USNM439006 (1 male), collected with the holotype; USNM439007 (2 males, 1 female), 10 April 2014, Morris Creek at the holotype location; MMNS80818 (4 males), 22 April 2014, Morris Creek at the holotype location.

  _Paratypes_: MMNS80819 (2 males, 4 females), 11 June 1999, Pumpkin Creek at County Road 266, Lafayette County, Mississippi, 12.5 km E of Oxford ; MMNS80820 (2 males, 5 females), 9 June 1999, Yellow Leaf Creek on private property, Lafayette County Mississippi, 6.5 km E of Oxford; MMNS80821 (1 male, 3 females), 10 April 2014, Gordon Branch at CR 121, Yalobusha County, Mississippi, 7.6 km ESE of Water Valley; MMNS80822 (1 female), 10 April 2014, Johnson Creek at County Road 436, Yalobusha County, Mississippi, 3.9 km S of Water Valley; MMNS80823 (4 males), 21 April 2014, Gordon Branch at CR 121, Yalobusha County, Mississippi, 7.6 km ESE of Water Valley; MMNS80824 (5 males, 1 female), 22 April 2014, Mill Creek at State Highway 315, Yalobusha County, Mississippi, 5.7 km WSW of Paris; MMNS80825 (3 males, 2 females), 12 October 2017, Splinter

Creek at State Highway 328, Lafayette County, Mississippi, 8.9 km N of Water Valley; USNM439008 (2 males, 4 females), 12 October 2017, Splinter Creek at County Road 348, Lafayette County, Mississippi, 10 km N of Water Valley; MMNS80826 (4 females, 1 juvenile), 9 June 1999, Yellow Leaf Creek on private property downstream of a confluence with an unnamed tributary, Lafayette County, Mississippi, 8.2 km E of Oxford and 16.3 km SSE of Abbeville; TU152109 (4 males, 6 females), 11 May 1988, un-named tributary to Taylor Creek at Old Taylor Road, Lafayette County, Mississippi, 2.7 km N of Taylor; TU3116 (5 males, 6 females), 24 May 1952, Pumpkin Creek at State Highway 6, Lafayette County, Mississippi, 12 km ESE of Oxford; TU155210 (6 males, 15 females), 14 June 1989, un-named tributary to Otoucalofa Creek at State Highway 32, Yalobusha County, Mississippi, 4.2 km SSE of Water Valley; TU155225 (1 males, 2 females), 15 June 1989, Gordon Branch at County Road 121, Yalobusha County, Mississippi, 7.8 km ESE of Water Valley.

## Diagnosis

*Etheostoma faulkneri* is one of at least 27 species, undescribed forms, or Evolutionary Significant Units of snubnose darters in the unranked clade Adonia (sensu *Near et al., 2011*; see *Blanco, 2001*; *Brogdon & Tabit, 2003*; *Boschung & Mayden, 2004*; *Gabel, 2007*; *Kozal et al., 2017*) (Table S2) and is indicated as the sister species of *E. raneyi* (*Sterling et al., 2020*). Consistent with other Adonia snubnose darters, *E. faulkneri* lack a frenum and usually have several long, thin teeth on the vomer (*Suttkus, Bailey & Bart Jr, 1994*; *Kozal et al., 2017*). Modal lateral line scale counts differ between *E. faulkneri* (45) and *E. raneyi* (47). *Etheostoma faulkneri* males and females have shorter snouts and heads and more robust bodies relative to *E. raneyi* and are longer. Compared with *E. raneyi*, male *E. faulkneri* usually have a smaller or no clear window in the blue anal fin and more extensive blue pigment on the anal fin, second dorsal fin, procurrent rays of the caudal fin, the cheek, opercle, and tip of the snout during the spawning season. *Etheostoma faulkneri* also lack orange pigment in the anal fin which is present in a small proportion (17.7%) of male *E. raneyi*, and the black pigment on the opercle, cheek, chin, branchiostegal rays, pelvic fins, second dorsal fin, and caudal fin is generally less dense and less extensive for *E. faulkneri* than for *E. raneyi*. This results in spawning male *E. faulkneri* usually appearing brighter blue than *E. raneyi*, which appear duskier (Fig. 4). *Etheostoma faulkneri* can be distinguished from *E. raneyi* by fixed allele differences at 12 loci on the mitochondrial *cytb* gene and one difference and two deletions on the nuclear *S7* gene (*Sterling et al., 2020*).

## Description

Maximum length for *E. faulkneri* is a 64 mm SL male sampled from Morris Creek. For females, three 49 mm SL individuals were each sampled from Yellow Leaf, Johnson, and Morris creeks. Maximum SL for *E. raneyi* is 57 mm for males and 50 mm for females (K Sterling, pers. obs., 2011–2019).

The lateral line is complete with 41–51 scales (mode = 45) (Table 3). The last scale is sometimes unpored. Transverse scales range from 11–15 (mode = 13) (Table 4), and scales around the caudal peduncle range from 16–19 (mode = 17) (Table 5). Dorsal fin
spines range from 9–11 (mode = 10), second dorsal fin rays range from 9–11 (mode = 11) (Tables 6–7), anal fin rays range from 6–8 (mode = 7), and pectoral fin rays range from 13–15 (mode = 14) (Tables 8–9). The belly, nape, cheek, and opercle are scaled, and the breast is usually naked but frequently has several scales, especially for large individuals. The prepectoral area is often scaled, but for some individuals scales are limited in extent.

The spinous dorsal fin of male *E. faulkneri* has a bright turquoise margin, an incomplete medial band of bright red in the posterior 3–5 interradial membranes that fades anteriorly into a metallic gold color best seen against a black background but appears as beige patches of pigment against light backgrounds (described as pale cream in *Suttkus, Bailey & Bart Jr, 1994*). A band of black that is usually incomplete is basal to the red and gold band and is present in the anterior 1–6 fin membranes. During the spawning season, this black band is often suffused or overlain with turquoise pigment. The soft dorsal fin margin has a diffuse posterior patch of turquoise; the turquoise pigment becomes limited to the fin rays anteriorly and does not compose a band of color (we did not observe a band of turquoise in the soft dorsal fin as described in *Suttkus, Bailey & Bart Jr, 1994*) for either *E. raneyi* or *E. faulkneri*. A bright orange medial band of color in the interradial membranes extends from the posterior edge of the soft dorsal fin and fades and becomes thinner anteriorly through 5–8 membranes. A basal patch of black pigment often occurs where the fin meets a dorsal saddle. Black pigment is frequently present in the interradial membranes of the fin but is usually less dense and less extensive in area compared with *E. raneyi*, though in some individuals the black pigment is well developed.

The anal and pelvic fins are bright turquoise in nuptial males. A small, thin, clear to creamy medial window in the anal fin is often present in the posterior 2–3 interradial membranes. Turquoise pigment is well developed from the margin of the anal fin to the belly. In contrast, *E. raneyi* usually have larger windows that cover 3–4 membranes and are wider. In some individuals, black pigment is present in the membranes of the pelvic fins, which is usually denser and more extensive in *E. raneyi*. The pectoral fins of *E. faulkneri* are mostly devoid of pigment, but the rays have intense orange pigment proximal to the fin insertion. The caudal fin is mostly clear, but the procurrent rays of nuptial males are bright turquoise blue, and the primary rays may have some green, blue, or black pigment.

The background color of adult males is a warm cream color dorsally that becomes paler ventrally. Large males have intense dark orange pigment from the mid-ventral area up to the mid-lateral stripe that fades to a lighter, bright orange anteriorly. In some males, the orange is also present in the prepectoral area. In small males, the orange is reduced to a band of pigment along the side of the belly that does not extend to the mid-lateral stripe or to the anal fin. Two basicaudal spots of orange are almost always present. The mid-lateral stripe consists of a series of blotches bisected by a depigmented lateral line that becomes indistinct posteriorly. Blotches alternate between a dark brown to black color and a dark red color. The dark red blotches are usually elongate rectangles. The dark brown or black blotches are often the ventral portion of a y-shaped pattern that extends to the dorsal fins and is connected by the dorsal saddles, of which there are eight (see Fig. 4). However, these blotches are irregularly shaped and patterns are often irregular, especially anteriorly. The dark brown and black pigment becomes an irregular series of spots and blotches on the

head. A dark to dusky suborbital bar extending below the eye is usually present as is a dark pre-orbital stripe. The top of the head is the same dark brown as the dorsal saddles. In nuptial males, there may be a green to turquoise tint on the top of the head and anterior saddles as well as the tip of the snout. A clear, bright turquoise color is present on the pre-opercle, opercle, cheek, chin, branchiostegal rays, and breast. In some individuals, black spots of pigment are present in these areas which cause the bright turquoise to appear dusky, an effect more often seen in *E. raneyi*.

Females are more cryptically colored although the pigment patterns of the mid-lateral stripe and dorsal surface of the body are highly similar to the male. The background color is a neutral tan dorsally and becomes an unsullied bright white ventrally. The mid-lateral stripe is an alternating series of dark maroon to warm brown and dark brown to black blotches bisected by an unpigmented lateral line that becomes more indistinct posteriorly. The maroon or warm brown blotches are elongate rectangles, and the dark brown to black blotches are usually irregular in size and shape, sometimes connected to the dorsal saddles. Blotches become more irregular anteriorly and are small on the head and opercle. A dark to dusky pre-orbital stripe and suborbital bar usually are present. The belly is usually devoid of pigment, though large females may have some small spots of orange pigment. Two orange basicaudal spots are almost always present. In nuptial females, a light wash of turquoise may occur on the breast, tip of the snout, and top of the head. Fins are all mostly devoid of bright pigment with clear membranes and a series of elongate dark brown to black areas of pigment on the straw-colored dorsal and caudal fin rays. Occasionally, the red median band in the spinous dorsal fin of males is present in females though the color is much less intense and often appears as an indistinct series of spots in the most posterior 1–3 membranes. The pelvic, pectoral, and anal fins are usually clear though an indistinct wash of turquoise may be present in the anal fin.

*Distribution: Etheostoma faulkneri* is endemic to perennial headwater tributaries of the Y.R. drainage. To the east, distribution appears limited by the increased presence of intermittent streams in terrain with less relief and more extensive agricultural land (Fig. 1). Otoucalofa Creek, a large tributary to the Y.R., may have the largest connected population of *E. faulkneri* (*Sterling et al., 2012*; *Sterling, Warren Jr & Henderson, 2013*). The species is known from 33 sites in 20 streams; 11 streams are in the Otoucalofa Creek system (Table S3), and future sampling will likely yield *E. faulkneri* in several additional streams that appear to have suitable habitat. However, repeated sampling indicates that *E. faulkneri* are uncommon in most streams of occurrence and may even be extirpated from Smith Creek in Calhoun County (A. Carson, pers. comm., 2016–2017). The earliest known collection was by R.D. Suttkus on May 24, 1952 when 11 individuals were taken in Pumpkin Creek at State Highway 6, Lafayette Co., Mississippi (TU 3116).

## Habitat, biology, and population genetics

The biology and habitat use of *E. faulkneri* is apparently similar to that of *E. raneyi* (*Knight & Cooper, 1987*; *Suttkus, Bailey & Bart Jr, 1994*; *Sterling & Warren Jr, 2017*). A life history study of *E. faulkneri* was conducted at Morris Creek (*Johnston & Haag, 1996*), results of which are consistent with the literature for other clade Adonia snubnose darters (*Carney &*

*Burr, 1989*; *Khudamrongsawat & Kuhajda, 2007*; *Barton & Powers, 2010*). In our experience in the field, *E. faulkneri* are most commonly sampled from less-disturbed small streams with strong perennial flow and plentiful instream cover (wood or hard clay riffles). In streams lacking cover because of channelization, incisement, and sedimentation, the species is often collected from the rubble, wood pilings, and human refuse associated with bridge crossings and is sparse elsewhere. The first known description of habitat use by *E. faulkneri* (specific to the Otoucalofa Creek watershed) indicates that the species is usually found in densely canopied first-order streams that have not been clear-cut and are associated with hard-clay riffles (*Knight & Cooper, 1987*). Genetic studies indicate that *E. faulkneri* has lower genetic diversity relative to *E. raneyi*, likely suffers from decreased gene flow because of anthropogenic habitat destruction and fragmentation, and effective population sizes are likely dangerously low (*Powers & Warren Jr, 2009*; *Sterling et al., 2011*; *Sterling et al., 2012*; *Sterling et al., 2020*)

## Conservation management

*Etheostoma faulkneri* is restricted to one headwater drainage in the upper Yazoo River basin with a total area of only about 1,500 km$^2$ (relative to about 3,200 km$^2$ for *E. raneyi*) and is distributed almost entirely on private lands (*Sterling, Warren Jr & Henderson, 2013*). As described earlier, the species has low genetic diversity and is nowhere abundant outside of small stream reaches associated with some road crossings. Streams in western Tennessee and northern Mississippi are among the most altered in the United States and suitable habitat for several imperiled snubnose darters including *E. faulkneri* is lacking (*Shields Jr, Knight & Cooper, 1995*; *Warren Jr, Haag & Adams, 2000*; *Keck & Etnier, 2005*; *Fore et al., 2019*). Urban development in the Y.R. drainage is increasing rapidly in association with the explosive growth of the city of Oxford and, to a lesser extent, Water Valley, MS. As a result, *E. faulkneri* faces increasing habitat degradation through time across most of its range. However, the Otoucalofa Creek watershed (Fig. 1) is not yet under the direct threat of increased rates of development and appears to hold the largest connected population of *E. faulkneri* (*Sterling et al., 2012*; *Sterling, Warren Jr & Henderson, 2013*). Land use is still mostly dominated by timber plantations and row crop agriculture upstream of Water Valley. The United States Fish and Wildlife Service (USFWS) (Jackson, MS) and United States Department of Agriculture Natural Resources Conservation Service (NRCS) (Jackson, MS) have also taken steps to improve aquatic organism passage and riparian habitat in two streams in the Otoucalofa Creek watershed (*Natural Resources Conservation Service, 2020*; *USFWS, 2020*). Time possibly remains for the public to become aware that a unique part of their natural heritage swims in the streams in their backyards, on their farms, and the lands they hunt. Increased awareness may help to protect enough habitat for *E. faulkneri* to persist in the Otoucalofa Creek watershed over the coming century.

## Etymology

We have named the species *Etheostoma faulkneri* to honor the great writer and Nobel Laureate William C. Faulkner (1897–1962), a native of the Oxford, Mississippi area who was also an avid hunter and fisher. The landscape was an important theme in many of his

works, and the actions of his characters were often influenced by the lands and streams surrounding his fictional Jefferson, Mississippi, including the Yocona River, which he renamed the Yoknapatawpha.

## DISCUSSION

*Etheostoma faulkneri* masqueraded as *E. raneyi* for decades mostly because of the lack of distinctive color differences in nuptial males. Even so, patterns of distribution (Fig. 1) and quantitative differences in lateral-line scale counts (Table 3), male nuptial color (Figs. 4 and 5), length (Table 10), and morphology (Table 10; Fig. 6) between individuals sampled from the Y.R. and L.T.R., as well as population genetic and phylogenetic data (*Powers & Warren Jr, 2009*; *Sterling et al., 2012*; *Sterling et al., 2020*) constitute multiple lines of evidence that support the recognition of *Etheostoma faulkneri* as a separately evolving metapopulation and valid species under the unified species concept (*de Queiroz, 2007*).

The subtle differences in male nuptial color, meristics, and morphology indicated by our results are similar to those found between other recently diverged pairs of snubnose darters (*Suttkus, Bailey & Bart Jr, 1994*; *Bauer, Etnier & Burkhead, 1995*; *Powers & Mayden, 2003*). For example, diagnostic meristic characters between *E. cyanoprosopum* and *E. zonistium* are limited to modal lateral-line and transverse scale counts and male nuptial color differences are limited to the presence of a red ocellus in the first interradial membrane in *E. zonistium*, a character that is not universal among all nuptial males (K. Sterling, pers. obs., 2009–2020) (*Kozal et al., 2017*). Another recently diverged species pair, *E. cervus* and *E. pyrrhogaster*, show only a modal difference in lateral-line scale counts, a tendency for nuptial males of one species to have green pigment on the ventral portion of the head but the other species does not, and quantitative differences in two truss measures (*Powers & Mayden, 2003*). Though distinct differences in male color among closely related taxa are sometimes described (*Boschung, Mayden & Tomelleri, 1992*), subjective descriptions of color differences are likely to vary according to the source (e.g., *Boschung, Mayden & Tomelleri, 1992*; *Suttkus & Bailey, 1993*). In some cases, described differences in male color are less than distinct (e.g., *E. raneyi*, *E. ramseyi* and *E. lachneri*; *Suttkus, Bailey & Bart Jr, 1994*) across the range of individual variation within species (see *Ulocentra* key in *Boschung & Mayden, 2004*). Qualitative differences in meristic and morphological characters are rare among Adonia snubnose darters and are limited to counts of branchiostegal rays which separate *E. coosae* from the rest of the clade (except for *E. scotti*, *Bauer, Etnier & Burkhead, 1995*). Our results are consistent with these observations and do not indicate any qualitative differences between *E. raneyi* and *E. faulkneri*.

Because pigment patterns, meristics, and body shape cannot reliably distinguish individuals from the L.T.R. and Y.R., and the only diagnosable characters that qualitatively distinguish between *E. faulkneri* and *E. raneyi* are genetic (*Powers & Warren Jr, 2009*; *Sterling et al., 2020*), it is reasonable to categorize *E. faulkneri* as a cryptic species. Cryptic species can occur through several mechanisms that are not mutually exclusive: evolutionary convergence, recent divergence, and phylogenetic niche conservatism (*Fišer, Robinson & Malard, 2018*). Evolutionary convergence is clearly not a factor. However, the estimated

time of separation between *E. faulkneri* and *E. raneyi* (<1 my) is recent (*Sterling et al., 2020*), and niche conservatism is apparent between the two species (*Johnston & Haag, 1996*; *Sterling & Warren Jr, 2017*; *Ruble, Sterling & Warren Jr, 2019*) and other snubnose darters as well (*O'Neil, 1981*; *Carney & Burr, 1989*; *Hicks, 1990*; *Khudamrongsawat & Kuhajda, 2007*; *Barton & Powers, 2010*; *Hubbell & Banford, 2019*). *Etheostoma faulkneri* and *E. raneyi* exist in adjacent drainages with little latitudinal gradient, no elevational gradient, and no clear differences in surface geology, water chemistry, or aquatic communities. Their Eltonian and Grinnellian niches appear to be identical in all respects but one: there is a small difference in stream size and gradient between streams that the two species inhabit. Though this difference has likely influenced the subtle morphological divergence that we detected, stabilizing selection for their ancestral niche across a short time span since separation has clearly prevented the evolution of readily observable divergent morphological, meristic, or color characters.

Cryptic diversity is common among diverse taxa (*Pfenninger & Schwenk, 2007*; *Adams et al., 2014*; *Fennessy et al., 2016*), including freshwater fishes in North America (*Egge & Simmons, 2006*; *April et al., 2011*). Among darters (Etheostomatinae), cryptic diversity is especially common (*April et al., 2011*) and is linked to niche conservatism, vicariant events and allopatric distributions, and relatively stable habitat conditions over geologic time-scales (*Bauer, Etnier & Burkhead, 1995*; *Page, Hardman & Near, 2003*; *Near & Benard, 2004*; *Hollingsworth Jr & Near, 2009*; *Kozal et al., 2017*). These conditions are consistent with the literature on *E. faulkneri* and *E. raneyi* and our results (*Johnston & Haag, 1996*; *Powers & Warren Jr, 2009*; *Sterling & Warren Jr, 2017*; *Sterling et al., 2020*), as they are for other Adonia snubnose darters (*Carney & Burr, 1989*; *Bauer, Etnier & Burkhead, 1995*; *Powers & Mayden, 2003*; *Khudamrongsawat & Kuhajda, 2007*; *Kozal et al., 2017*).

## CONCLUSIONS

The results we present here, existing genetic evidence (*Powers & Warren Jr, 2009*; *Sterling et al., 2012*; *Sterling et al., 2020*), and a growing acceptance among biologists that the presence of readily observable qualitative diagnostic characters are not necessary to describe biodiversity (*Egge & Simmons, 2006*; *de Queiroz, 2007*; *Fišer, Robinson & Malard, 2018*), supports the recognition and description of *Etheostoma faulkneri*, the Yoknapatawpha Darter, as an independently evolving metapopulation lineage and valid species under the unified species concept (*de Queiroz, 2007*). The description of *E. faulkneri* represents an increase in the accuracy of our understanding of freshwater fish evolution and diversity, which is the foundation for research and conservation efforts for the many imperiled freshwater fishes of the southeastern United States.

## ACKNOWLEDGEMENTS

We thank the many people who contributed to this project by assisting in the field and lab, sharing ideas and information, and providing logistical support: S Adams, Z Barnett, H Bart, S Bingham, M Bland, A Carson, W Haag, G Henderson, J Hubbell, C Jenkins, G

McWhirter, S Nielsen, B Noonan, C Sabatia, J Schaefer, C Smith, S Smith, B Sterling, W Sterling, and M Wagner.

### Funding
This work was supported by the USDA Forest Service, Southern Research Station, Center for Bottomland Hardwoods Research. The funders had no role in study design, data collection and analysis, decision to publish, or preparation of the manuscript.

### Grant Disclosures
The following grant information was disclosed by the authors:
USDA Forest Service, Southern Research Station, Center for Bottomland Hardwoods Research.

### Competing Interests
The authors declare there are no competing interests.

### Author Contributions
- Ken A. Sterling conceived and designed the experiments, performed the experiments, analyzed the data, prepared figures and/or tables, authored or reviewed drafts of the paper, and approved the final draft.
- Melvin L. Warren, Jr. conceived and designed the experiments, analyzed the data, authored or reviewed drafts of the paper, and approved the final draft.

### Animal Ethics
The following information was supplied relating to ethical approvals (i.e., approving body and any reference numbers):

This study was conducted with the approval of the University of Mississippi, Department of Biology, IACUC Committee (protocol 09-027).

### Field Study Permissions
The following information was supplied relating to field study approvals (i.e., approving body and any reference numbers):

Collection permits were issued to us from the Mississippi Museum of Natural Science (Mississippi Department of Wildlife Fisheries and Parks), Jackson, MS (permit numbers: 0604091, 0513101, 0624112, 0622122, 0602132, 0610142, 0715163, 1010173).

### Data Availability
The raw data are available in a Supplemental File and at MorphoBase: Project 3712: Ken A. Sterling, Melvin L. Warren Jr. 2020. Description of a New Species of Cryptic Snubnose Darter (Percidae: Etheostomatinae) Endemic to North-Central Mississippi. PeerJ. (In Press). Project DOI: 10.7934/P3712, http://dx.doi.org/10.7934/P3712.

The data at MorphoBank was used in morphological analyses and consists of photography files, output from TPS software which converts landmarks on photographs to Cartesian coordinates, and output from MorphoJ software that analyzed the output from TPS.

Specimens from our collections were deposited with the National Museum of Natural History (Washington, D.C.) (USNM) or the Mississippi Museum of Natural Sciences (Jackson, MS) (MMNS). We obtained additional specimens from Tulane University Biodiversity Research Institute (Belle Chasse, LA) (TU): USNM439004 (1 male); USNM439005 (1 female); USNM439006 (1 male); USNM439007 (2 males, 1 female); USNM439008 (2 males, 4 females); MMNS80818 (4 males); MMNS80819 (2 males, 4 females); MMNS80820 (2 males, 5 females); MMNS80821 (1 male, 3 females); MMNS80822 (1 female); MMNS80823 (4 males); MMNS80824 (5 males, 1 female); MMNS80825 (3 males, 2 females); MMNS80826 (4 females, 1 juvenile); TU152109 (4 males, 6 females); TU3116 (5 males, 6 females); TU155210 (6 males, 15 females); TU155225 (1 males, 2 females).

### New Species Registration

The following information was supplied regarding the registration of a newly described species:

Publication LSID: urn:lsid:zoobank.org:pub:6C5BEC69-22A1-4008-84D3-CA62719AE152

Etheostoma faulkneri LSID: urn:lsid:zoobank.org:act:B9FE97A8-A86C-41A9-9CD6-2929362E0C22.

### Supplemental Information

Supplemental information for this article can be found online at http://dx.doi.org/10.7717/peerj.9807#supplemental-information.

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
