# Peer review of "Description of a new species of cryptic snubnose darter (Percidae: Etheostomatinae) endemic to north-central Mississippi"

_PeerJ, doi:10.7717/peerj.9807_

## Round 0.1 · original submission · Minor Revisions

Your manuscript has now been seen by two reviewers who consider the work to be a solid and important contribution. Please address each of their comments - they both point out a number of ambiguities and issues regarding clarity that need to be addressed.

Specifically, I would strongly suggest that the species diagnosis be pulled from the description and given a separate subheading.

Reviewer 1 ·

Basic reporting

Animal usage and permitting descriptions are presented and seem appropriate. Well written and depth of literature looks good. Figures seem fine.

New species:
New species name is not in title, but it is in the abstract.
New species has been entered into ZooBank according to methods. However I was unable to locate it with the link provided. Presumably because this paper is not accepted yet.
I honestly can judge whether it meats ICZN standards. However they do provide a generally recognized (modern) species definition that they are applying.

Experimental design

This paper is essentially a follow up to a just published genetics paper (DOI 10.7717/peerj.9014) where they suggest that there is a cryptic species present. Here, the authors present a detail phenotypic analysis to quantify and determine support for the species diagnosis and provide a taxonomic description. There was no detected flaws or concerns with the methodology or design of the study.

Validity of the findings

I think the findings are valid, however there is always the nagging feeling that it is too easy to pick a species definition that fits the data.
One thing that I struggled with was 1) the lack of (even a brief) description of what the unified species definition is (criteria), and in the discussion you don't clearly link your findings to said criteria. This would be valuable for some readers.

Additional comments

Line 453-458 you write "..quantitative differences in lateral line scale..color... length...and morphology...support the recognition of ... valid species"
Line 470 quantitative differences we detected are similar to those found for other species of snubnose darters"
If the timing of divergence is recent relative to other snubnose darters sibling or sister taxa, are you saying that faulkneri and raneyi display phenotypic divergence comparable to other examples of more deeply diverged taxa? If so, perhaps a more specific reference to examples would be warranted in the discussion, as this would help to clarify/strengthen your argument for species recognition.

Line 98- closed parenthesis is italicized at end of sentence.

Line 455 remove space after "Figure 6"

RE: Table S1. "Estimated stream gradients and watershed area for Yazoo Darters in the Yocona (Y.R.) and Little Tallahatchie (L.T.R.) River drainages."
Should be 'Yocona River and Little Tall.. River drainages' - or - 'Yocona and Little Tall.. river drainages' (where river is lower case in reference to both names)

Reviewer 2 ·

Basic reporting

Custom checks:
Ethical treatment of organisms covered under University of Mississippi IACUC (protocol 09-027).
The experiments (capture and photographing of fishes) were necessary to the study, however, the treatment of the photographed fishes is not stated (See ED below).
Appears to be a discrepancy in field collection permits, and this is probably due to omission of information in the text. In lines 54-56 it states that field collections were made using permits for the years 2009-2014, however, line 65 states that collections for photographs were made in 2017-2018. (See comments by line below).

Basic Reporting:
In general, clear, concise and well composed, however, there are some areas of ambiguity due to omission of information, awkward sentence structure or information not being organized in standard scientific format. The paper has an excellent foundation based on pertinent citations. One area that poses the greatest difficulty is the figures and tables. They are a bit of a mess when it comes to captions, legends and even omissions or correct information (e.g. Figure 1). I will list the figures and table problems in a subsequent list. Unfortunately, my computer would not open the raw data file(s).

Experimental design

Experimental Design:
This appears to be an original work based on a foundation of earlier endeavors. The subject is one that is well founded and in dire need of discussion. It is a very important addition to our understanding of diversity of freshwater fishes. The overall design is sound, however, there could (should) be additional statistical analyses performed. There is some confusion regarding methods. Once again, generally good, but some important details are lacking (e.g. Where the fishes that were photographed for nuptial coloration evaluation, anesthetized and released, or were they euthanized and preserved?). More specifics and their

Validity of the findings

Validity of Findings:
I was unable to assess the raw data. But there are minor inconsistencies in data description. Some of which may stem from inadequate captioning on figures and tables. This is a problem that was consistent for all tables and figures. Most importantly, their conclusion that Etheostoma faulkneri is an undescribed species confused with E. raneyi is well supported by their results.

It is a very important paper increasing our understanding and accounting of biodiversity in the southeastern U.S. and worldwide. Also, it is imperative for the conservation of this species.

Additional comments

General Comments:
A sound piece of research presented in this manuscript is deserving of Acceptance for publication with moderate revision which includes minor composition issues, organization and a need for further statistical testing (Calculation of 95% Cis are not sufficient to determine significant difference. Statistical analysis resulting in p-values is in dire need).
Below I list potential edits or issues line by line and for all figures and tables.

Lines:
27: Follow Upper Gulf Coastal Plain with abbreviation (UGCP).

34: Same as line 27 follow with acronyms or abbreviations i.e. (LGCP) and (MAP)

37-41: What is the genetic distance between lineages that has be found and what model was used. May want to not include this here but rather in lines 321-322.

55-56: The collecting permit discrepancy noted above.

58-68: Male nuptial color photos of live fishes. Were the fishes preserved or were they released. Was MS222 or the like used. I found the answer in the Supplement, but it needs to also be stated in the manuscript text.

69-82: Experimental design fine except perhaps the limited number of participants “colleagues” (only 3)

70-71: what was the sample size of darters photographed.

89-93: Differences in modal counts from literature “adjusted”. How? Please explain how adjustments were determined. Were the same fishes (museum specimens) observed by different authors?

96: “28 pairs of points” – very ambiguous as to what represents a pair of points that was measured. See Table 10 referred to in Figure 2 (not anywhere in the text at this point).. This is like a treasure hunt trying to find pertinent information in ms. This information should not be buried in a figure caption. Table 10 should be Table 2, chronologically if mentioned in text instead of referenced in figure 2.

98: “checked “ replace with assessed

99: “showed” replace with had

99-101: Why was the data not fully treated statistically using something like ANCOVA, also PCA or its equivalent.

106: “so” change to thus

109: citations for traditional multivariate needed to justify use of “morphometric”

107-113: Justification to use “morphometric approach” deemed appropriate, however, the phrase “morphometric” is ambiguous because there are many morphometric approaches used by systematists. Possible edit “ a morphometric approach as first described by Bookstein (1991) and subsequently modified (Parsons et al., 2003…
107: Possibly abbreviate geometric morphometric analysis as GMA.

119: On figure 3 I can only find 18 landmarks.

137-142: no statistical testing of means, p-values.

150-151: In Supplement it states fishes were released, or were they part of your collections (line 148)?

144: Now using phase “morphological analysis”. What does this include Truss, GMA, etc…?

148-149: Brings up question once again of how your specimens were handled, killed/euthanized, preservation techniques…

172: See comments on figure (#4).

174: No p-values to determined for testing significant differences, no stats other than 95%CI; See comments on caption and figure (#5)

177: 17.7% of what, no sample size reported.

182: See comments on table 3

193: GMA previously called morphological analyses on line 144?

201-210: A point alone cannot be longer or shorter, needs reference to some other point; reword

205-217: Since a large percentage of individuals are mis-identified from wrong drainage or wrong sex a multivariate plot (e.g. PCA) would be a helpful figure to include

227 replace > with greater than

226: Cannot make this conclusion from the data presented. Need actual hydrologic data (i.e. discharge)

230: add Synonomy

275: insert mm after 41.2

250-268: Don’t think the parenthetical information is needed in a synonomy.

296, 299: Un-named, lower case, not a proper noun

308: (Table S2), S is not defined.

311: Lat line scale counts differ, are they significantly different?

320: remove “overall” at end of sentence

321-322: What is the genetic distance and model used.

323-326: New paragraph

328-329: Why is no “mode” listed for transverse scales and scales around caudal peduncle?

341-342: insert “margin” after dorsal fin and delete “on the fin margin”

343: replace “there is no” with “we did not observe a”

352: Not clear what you are talking about, I do not see it (fig 4)

371-372: On Fig 4 I do not see a dark or dusky suborbital bar.

386: is somewhat redundant with line 380

425: Did not read anything about genetic dversity

427-429: Awkward sentence

436-437: Spell out all first use of acronyms; we are not all Feds.

444-: Etymology Very nice!

509: ?

514: remove “the” before lab

Figures:

1. Colored lines are not consistent from legend to map

2. insert (1-27) after landmarks and remove “is shown”.

3. replace “shown” with “indicated by numbers 1-18”. And if someone sees this figure in greyscale they will not be able to find #16

4. Rearrange completely. Label male holotype as “a” and move photo to top position; female allotype as “c and move appropriately. Change “d” to “b” and “b” to “d” . Also, put 1cm on black bars. They look like something has been marked out.

5. sample sizes should be reported in main text. The colors of the bars should be more distinctive so that one viewing in grayscale can see the difference.

6 and 7. Move legend for rivers into text of caption. Replace “Output from” to “Results of”.

8. Put a before b in caption!


For figures and tables the captions should contain all information needed to understand the figure or table without having to refer to the text of the paper. Yours tend to be lacking in pertinent information and do not use “shows” or “shown”.

---

## Round 0.2 · accepted · Accept

The authors have done an excellent job revising the manuscript and have adequately addressed the comments made by me and both reviewers.